# Object-Level Pseudo-3D Lifting for Distance-Aware Tracking

Haoyuan Jin*
Zhejiang University,
Hangzhou, Zhejiang, China
jhyjhy@zju.edu.cn

Xuesong Xie*
Zhejiang University,
Hangzhou, Zhejiang, China
xuesongnie@zju.edu.cn

Yunfeng Yan†
Zhejiang University,
Hangzhou, Zhejiang, China
21210004@zju.edu.cn

Xi Chen
University of Hong Kong,
Hong Kong, China
chauncey0620@gmail.com

Zhihang Zhu
Zhejiang University,
Hangzhou, Zhejiang, China
22210044@zju.edu.cn

Donglian Qi
Zhejiang University,
Hangzhou, Zhejiang, China
qidl@zju.edu.cn

## Abstract

Multi-object tracking (MOT) is a pivotal task for media interpretation, where reliable motion and appearance cues are essential for cross-frame identity preservation. However, limited by the inherent perspective properties of 2D space, the crowd density and frequent occlusions in real-world scenes expose the fragility of these cues. We observe the natural advantage of objects being well-separated in high-dimensional space and propose a novel 2D MOT framework, "Detecting-Lifting-Tracking" (DLT). Initially, a pre-trained detector is employed to capture 2D object information. Secondly, we introduce a Mamba Distance Estimator to obtain the distances of objects to a monocular camera with temporal consistency, achieving object-level pseudo-3D lifting. Finally, we thoroughly explore distance-aware tracking via pseudo-3D information. Specifically, we introduce a Score-Distance Hierarchical Matching and Short-Long Terms Association to enhance accurate and robust association capability. Even without appearance cues, our DLT achieves state-of-the-art performance on MOT17, MOT20, and DanceTrack, demonstrating its potential to address occlusion challenges.

## CCS Concepts

• **Computing methodologies → Tracking**.

## Keywords

Media Interpretation, Multi-Object Tracking, Pseudo-3D Lifting, Distance-Aware Tracking

**ACM Reference Format:**
Haoyuan Jin, Xuesong Xie, Yunfeng Yan, Xi Chen, Zhihang Zhu, and Donglian Qi. 2024. Object-Level Pseudo-3D Lifting for Distance-Aware Tracking. In *Proceedings of the 32nd ACM International Conference on Multimedia (MM '24), October 28-November 1, 2024, Melbourne, VIC, Australia.* ACM, New York, NY, USA, 9 pages. https://doi.org/10.1145/3664647.3680783

*Both authors contributed equally to this research.
†Corresponding author.

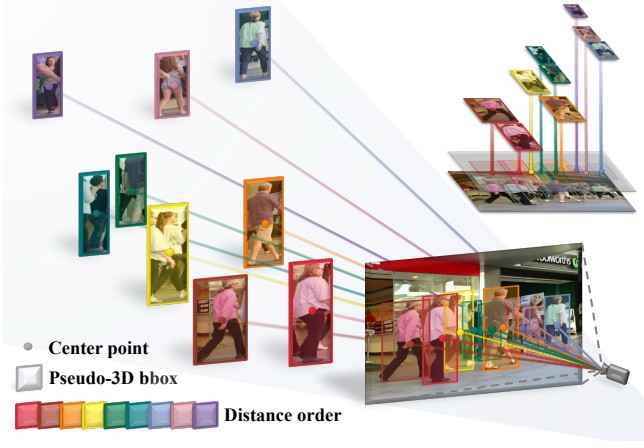

**Figure 1: The illustration of object-level pseudo-3D lifting. Objects that are densely distributed and occluded in a monocular 2D image are well-separated in pseudo-3D space.**

## 1 Introduction

Multi-object tracking (MOT) is a critical task for media interpretation, involving the spatial localization of objects and the temporal association to maintain consistent identities across consecutive video frames. This task supports a wide range of applications such as autonomous driving [33, 39] and intelligent surveillance [9, 36]. However, due to the complexity of real-world scenarios, including dense crowds and frequent occlusions, MOT with 2D images from a monocular camera remains a highly challenging topic.

Previous works [16, 38, 43] explore MOT extensively and gradually formulate an effective paradigm named tracking-by-detection (TBD). This paradigm divides the task into two sequential sub-tasks: detecting objects in each frame, followed by associating these objects across frames based on motion and appearance cues which are crucial for tracking performance. However, when faced with the occlusion problem, the confusion caused by the mutual influence of objects in 2D space makes the reliability of these clues significantly degraded. Several works [26–28, 43] address this issue by improving motion prediction, enhancing appearance representation, and designing matching strategies, yet they still suffer from the inherent limitations of the 2D perspective. In contrast, we recognize the natural advantages of matching in high-dimensional space, where

objects that occlude each other in 2D space are well-separated, shown in Figure 1. Therefore, we would like to ask: *How to achieve multi-object tracking in 3D space based solely on 2D images?*

In our work, we introduce an innovative 2D MOT framework, **"Detecting-Lifting-Tracking" (DLT)**, which decomposes the discussed problem into two questions: (i) *How to achieve the transformation from 2D space to 3D space?* (ii) *How to fully leverage 3D information for a comprehensive tracking process update?*

(i) Object-level pseudo-3D lifting. For spatial transformations without any sensor or various views, the most straightforward 3D lifting method is the use of depth estimators. DIP [17] and Quo Vadis [6] employ this method for reconstructing 3D scenes and providing inference bases for invisible objects, respectively. Despite the high computational cost for scene-level depth, depth estimators still suffer from low accuracy and are prone to background noise. In contrast, estimating the distances between objects and the camera serves as a pseudo-3D lifting method, realized through distance estimators such as DisNet [14], EM [45], and DistSynth [23]. These distance estimators focus on capturing object-level distance, disregarding the redundancy of background. This is consistent with the object-specific perception of MOT tasks because it is more crucial to know the object distance instead of the dense depth map for the entire scene [2]. Thus, we introduce a Mamba Distance Estimator (MDE) based on the Space-Time Mamba Model, integrating historical information to mitigate the impact of temporary occlusions.

(ii) Distance-aware tracking. DLT aims to efficiently utilize pseudo-3D distance information for tracking. Bearing such objectives, we revisit the conventional tracking process, dividing it into two main parts: a set-level matching strategy and an instance-level association schema. On one hand, *the matching strategy* is the focus of MOT tasks. ByteTrack [43] enhances it by decomposing objects into high- and low-score subsets based on confidence scores, which shows surprisingly effective outcomes. For discrete and sequential matching, we propose the Score-Distance Hierarchical Matching (SDHM) strategy. Specifically, to tackle the issue of confidence scores failing to robustly evaluate object existence in occluded scenes, we introduce the Score-based Hierarchizing incorporating an Occlusion Compensation Score based on distance-aware visibility ratios. Additionally, a noticeable density of objects still persists within high-score subset, we propose Distance-based Hierarchizing for a secondary hierarchizing on the distance dimension. On the other hand, *the association schema* emphasizes object-track correlations to verify identities. We present the Short-Long Terms Association (SLTA) schema, distinguishing the Short-term Association in adjacent frames impacted by dense crowds, from the Long-term Association addressing re-association challenges due to severe occlusion. For the Short-term Association, the Kalman filter [37] is standard for motion prediction and track update, leveraging IoU for correlation metrics [3, 38, 44]. Our DLT introduces a Pseudo-3D Adaptive Kalman Filter that expands the 2D Kalman filter into pseudo-3D space and adaptively modifies prediction and update noise with object-level distance. We also introduce a new Distance-weighted IoU, incorporating distance variation as an explicit clue. For the Long-term Association, prior works primarily utilize discriminative appearance features which are occlusion-sensitive clues [28, 34]. Additionally, Quo Vadis [6] attempts to incorporate trajectory forecasting [4] for non-linear prediction with multiple complex

sub-modules. In contrast, we introduce a Probabilistic Autoregressive Motion Model for modeling natural trajectory distributions in pseudo-3D space, facilitating long-term motion prediction.

By answering the two specific questions mentioned earlier, DLT achieves accurate and efficient tracking in pseudo-3D space, setting a new state-of-the-art performance on the MOT17, MOT20, and DanceTrack test sets. Our contributions are summarized as follows:

- We propose a novel *"Detecting-Lifting-Tracking"* (DLT) framework for 2D MOT that leverages the natural advantages of pseudo-3D space without any sensors or various views.
- We introduce a Mamba Distance Estimator, combined with historical information, for temporally consistent distance estimation, achieving *object-level pseudo-3D lifting*.
- We present a Score-Distance Hierarchical Matching strategy and a Short-Long Terms Association schema by making full use of the distance clue to realize *distance-aware tracking*.
- We conduct comprehensive benchmark evaluations and ablation studies, demonstrating the exceptional performance of DLT, as well as the contribution of each key component.

## 2 Related Work

*Tracking-by-Detection.* As the reliability of object detection increases [10], the tracking-by-detection (TBD) paradigm gradually becomes the mainstream method in MOT tasks [3, 43]. TBD typically utilizes pre-trained detectors to obtain object bboxes, focusing on the matching process. SORT [2] and DeepSORT [38] are two of the most classic methods, employing Kalman filters for motion prediction and IoU for correlation measurement as the basis for the Hungarian algorithm. Despite the emergence of numerous advanced methods, challenges such as crowd density and occlusions in real-world scenarios still limit tracking performance. To address this issue, various solutions have been proposed, such as some methods employing carefully designed appearance modules to provide multi-modal cues [27, 28], and others introducing more reasonable motion prediction modules [6, 21, 26]. Differently, we consider a more direct and effective method, namely conducting pseudo-3D lifting and combining distance clues to optimize the matching process, thereby realizing a novel "Detecting-Lifting-Tracking" framework.

*Distance estimation.* Distance estimation and depth estimation are two distinct tasks, although both are based on 2D images without additional sensors or multiple viewpoints. Distance estimation solely aims to obtain the distance information of an object from the camera [19]. In contrast, depth estimation seeks to acquire the depth information for all pixels in a scene [42]. Classical methods of distance estimation achieve this by regressing the relationship between an object's geometric shape and its distance [22, 29]. The first attempt at a deep learning method estimates an object's distance based on the width and height of a given bbox using a Support Vector Machine Regressor (SVR) [12]. Similarly, DistNet [14], also a bbox-based method, does not employ image feature learning, resulting in significantly noticeable error noise. EM [45] combines ResNet and RoI pooling to reconstruct object-level representations for distance estimation. Building on EM, DistSynth [23] adds a real-time model and an FPN branch. Considering that distance estimation is critical for object-level pseudo-3D lifting, we design a Mamba Distance Estimator (MDE) based on the State Space Model [13],

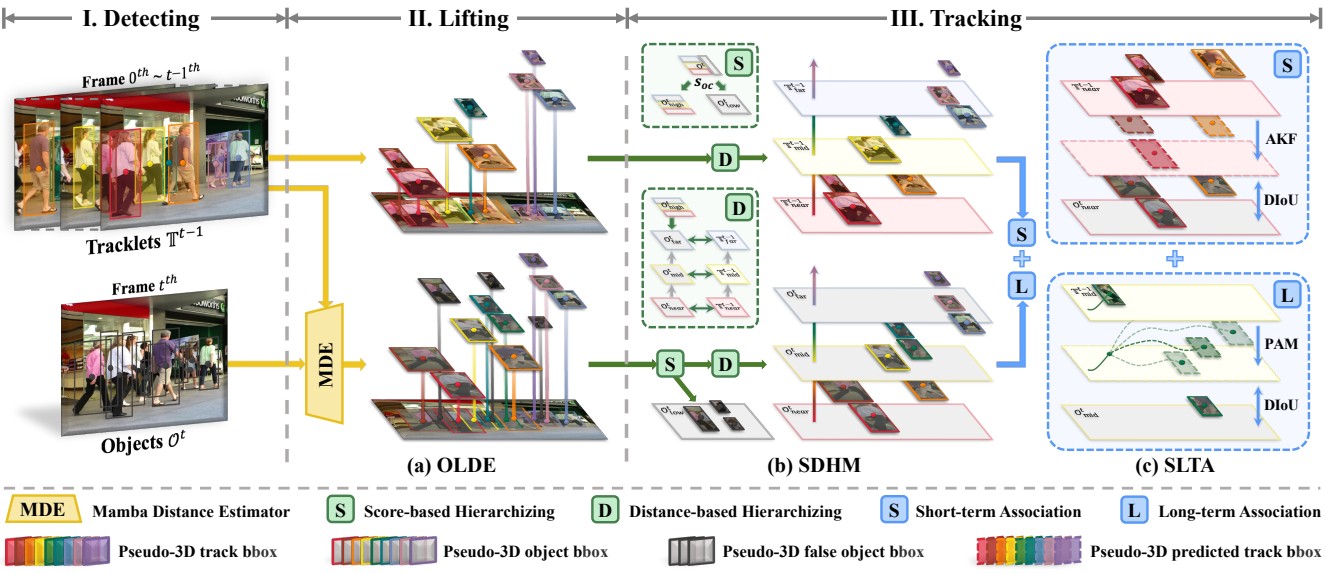

**Figure 2: Overview of our "Detecting-Lifting-Tracking" (DLT) framework. I. Detecting: capturing 2D object bboxes in the current frame by the detector. II. Lifting: utilizing *Object-Level Distance Estimation* (OLDE) with Mamba Distance Estimator to generate pseudo-3D information for each object. III. Tracking: achieving distance-aware tracking through the *Score-Distance Hierarchical Matching* (SDHM) combined with *Short-Long Terms Association* (SLTA). The SDHM incorporates a Score-based Hierarchizing, based on the Occlusion Compensation Score, and further implements a Distance-based Hierarchizing for high-score subsets. The SLTA comprises a Short-term Association between two adjacent frames by a Pseudo-3D Adaptive Kalman Filter with Distance-weighted IoU and a Long-term Association for re-association through a Probabilistic Autoregressive Motion Model.**

incorporating historical spatiotemporal information to mitigate the effects of occlusion and enhance estimation accuracy.

*Tracking with 3D information.* Our work focuses on utilizing 3D information for 2D MOT tasks, differing from 3D MOT [25], in that it inputs and outputs 2D bboxes, with the tracking process executed in 3D space. There are few such works, among which DIP [17] stands as a pioneering effort. It employs depth maps obtained from a depth estimator for inferring the presence of objects, thereby enhancing the capability to handle occlusions. Another notable work is Quo Vadis [6], which merges depth maps and segmentation results using homography transformations to generate bird's-eye view scene maps, thereby introducing trajectory forecasting to alleviate the influence of long-term object occlusion. While these works somewhat improve the handling of occlusions, they are very limited. The underlying reason is the insufficient utilization of 3D information. Hence, we comprehensively revisit the tracking process, implementing depth-aware enhancements in both matching strategy and association schema to considerably mitigate the inherent sensitivity of 2D MOT methods to occlusions.

## 3 Methodology

We propose an innovative **"Detecting-Lifting-Tracking" (DLT)** framework, as illustrated in Figure 2, which has three main stages: i) detecting to obtain 2D object bboxes, ii) lifting to pseudo-3D space with *Object-Level Distance Estimation* (OLDE) by Mamba Distance Estimator, iii) tracking through *Score-Distance Hierarchical Matching* (SDHM) with *Short-Long Terms Association* (SLTA).

### 3.1 Object-Level Distance Estimation

In 2D MOT tasks, pixel-wise depth estimators provide an overly redundant amount of information for the entire scene. Our focus shifts toward Object-Level Distance Estimation (OLDE), which obtains the distances between specific objects and the camera. The primary challenges include: (i) previous estimators rely on single-frame images without temporal consistency, making them susceptible to occlusions, and (ii) it becomes even more complex when dealing with small objects, necessitating higher precision. To address these challenges, we introduce a Mamba Distance Estimator (MDE).

*Space-Time Mamba.* We introduce the Space-Time Mamba (ST-Mamba) to estimate distances from multi-frames, leveraging the inter-frame context instead of just 2D spatial information to achieve a smoother estimation. The Mamba consists of State Space Models (SSMs) [13], which map a 1D function or sequence $x(t) \in \mathcal{R}^L \mapsto y(t) \in \mathcal{R}^L$ through a hidden state $h(t) \in \mathcal{R}^N$. SSMs are often represented in a discretized version as follows:

$$\overline{\mathbf{A}} = \exp(\Delta \mathbf{A}), \tag{1}$$

$$\overline{\mathbf{B}} = (\Delta \mathbf{A})^{-1} [\exp(\Delta \mathbf{A}) - \mathbf{I}] \cdot \Delta \mathbf{B}, \tag{2}$$

$$h_t = \overline{\mathbf{A}} h_{t-1} + \overline{\mathbf{B}} x_t, y_t = \mathbf{C} h_t, \tag{3}$$

where a timescale parameter $\Delta$ to transform the continuous parameters $\mathbf{A}, \mathbf{B}$ to discrete parameters $\overline{\mathbf{A}}, \overline{\mathbf{B}}$. To adapt the vanilla Mamba for multi-frame input, we introduce the spatiotemporal selective scan, as illustrated in Figure 3. Specifically, we unfold patches of each

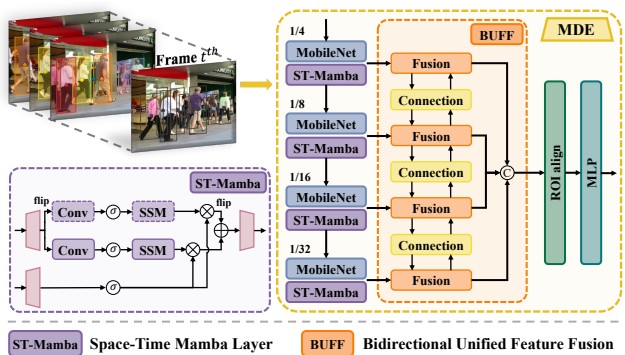

Figure 3: Structure of the Mamba Distance Estimator (MDE). MDE takes multiple frames as input and outputs an estimated distance. The lightweight MobileNet v3 extracts intra-frame features, while the ST-Mamba captures inter-frame representations. The BUFF module then integrates multi-scale features using both top-down and bottom-up approaches.

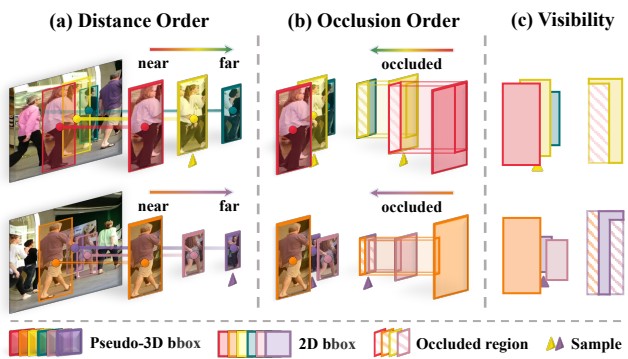

Figure 4: The illustration of the distance-aware object visibility analysis. Under the assumption that farther objects are occluded by nearer ones in a 2D perspective, the order of distance and occlusion can be easily mapped. The visibility is determined by assessing the ownership of overlapping areas.

frame along rows and columns into sequences and then concatenate the frame sequences to constitute the long sequences $X \in \mathcal{R}^{C \times THW}$. Then, we parallelly proceed with scanning along the forward and backward directions to explore inter-frame dependency. To balance estimation speed and accuracy, we insert Space-Time Mamba layers at the end of each stage in pre-trained MobileNet v3 [15], which extracts intra-frame features and fuses inter-frame information.

*Bidirectional Unified Feature Fusion.* To enhance accuracy for small objects, we introduce the Bidirectional Unified Feature Fusion (BUFF) module ensuring seamless feature fusion across scales. The BUFF module consists of Fusion and Connection components. The Fusion block is composed of several convolutional layers. We concatenate two input feature maps $X_1$ and $X_2$, which are fed into both branches. One branch compresses channels through a $1 \times 1$ Conv, while the other branch extracts features using a $1 \times 1$ Conv and Dilated Reparam Blocks [7] (DRepBlocks), ultimately the fused features $Y$ are obtained through an element-wise addition:

$$X = \text{Concatenate}(X_1, X_2), \quad (4)$$

$$Y = \text{Conv}_{1 \times 1}(X) + \text{DRepBlocks}(\text{Conv}_{1 \times 1}(X)). \quad (5)$$

The Connection module comprises upsampling and downsampling layers. In practice, the BUFF starts with top-down encoding for global features and then proceeds with bottom-up encoding to retain detailed features of small objects.

The final fused features are sent to an ROI align layer to extract object-level vector features $X_v \in \mathcal{R}^{N \times C'MM}$, where $N$ and $M$ denotes the number of objects and window dimension of the RoI align, respectively. These vectors are then transformed by an MLP layer to generate Gaussian distribution parameters $\hat{\mu}$ and $\hat{\sigma}^2$ for distance, ensuring a degree of uncertainty. Given ground truths $\mu$, the proposed MDE is optimized via Gaussian Negative Log Likelihood (GNLL) loss, denoted as follows:

$$\text{Loss}_{\text{GNLL}} = \frac{1}{2}\left[\log\left(\hat{\sigma}^2\right) + \frac{(\hat{\mu} - \mu)^2}{\hat{\sigma}^2}\right]. \quad (6)$$

## 3.2 Score-Distance Hierarchical Matching

The concept of hierarchical matching, due to its discrete and orderly attributes, has been proven to enhance tracking performance, yet there remains room for improvement: i) alleviating the hierarchical strategy sensitivity to occlusions, and ii) introducing new hierarchical methods to address occlusions further. Accordingly, we propose *Score-Distance Hierarchical Matching* (SDHM), which comprises *Score-based Hierarchizing* and *Distance-based Hierarchizing*.

*Score-based Hierarchizing.* As a vanilla method, ByteTrack [43] decomposes objects into two subsets based on confidence scores: high-score subset $O_{\text{high}}^t$ and low-score subset $O_{\text{low}}^t$. The $O_{\text{high}}^t$ is prioritized for matching. However, the confidence scores of some positive objects are reduced due to occlusions, resulting in putting into the $O_{\text{low}}^t$ and losing the privilege of priority matching. Thus, we propose an *Occlusion Compensation Score* ($s_{oc}$) to reflect the existence of objects enabling more robust *Score-based Hierarchizing*.

The visibility ratio of an object can, to a certain extent, reflect the impact of occlusions on the confidence score. However, this metric is not readily obtainable. For overlapping areas, complex methods such as comparing feature similarities are required to determine their ownership. Fortunately, based on the distance information, the overlapping area belongs to the nearer object. Therefore, for a specific object $o_a^t$ in $O^t$, its visibility ratio can be defined as:

$$r_{o_a^t} = 1 - \sum_{n \neq a}^{N_o^t} \frac{S_{I_{(o_n^t, o_a^t)}}}{S_{o_a^t}} \cdot H(d_{o_n^t} - d_{o_a^t}), \quad (7)$$

where $I_{(o_n^t, o_a^t)}$ is the overlapping area. $S$ represents the area of a polygon. $H(x)$ is the Heaviside function.

The $s_{oc}$ compensates the confidence score based on the visibility ratio. $O^t$ can be divided into two occlusion-insensitive subsets by setting $s_{oc}$ threshold. $O_{\text{high}}^t$ containing most of the positive objects, is prioritized for matching. The $s_{oc}$ is expressed as follows:

$$s_{oc} = \ln\left[e(2 - r)^\alpha\right] \cdot s, \quad (8)$$

where $s$ is the confidence score. $\alpha$ is the compensation degree.

*Distance-based Hierarchizing.* In 2D images, the positional overlap due to occlusion leads to mutual interference. Even with Score-based Hierarchizing, a noticeable density of objects persists within the $O_{high}^t$. However, by introducing distance information, their positions can be easily separated. Consequently, within the $O_{high}^t$, we further introduce *Distance-based Hierarchizing*.

For the $t^{th}$ frame, tracking involves bipartite graph matching between the objects set $O^t$ and the tracklet set $\mathbb{T}^{t-1}$ of the former frame. The $O_{high}^t$ can be further divided into three subsets based on the distance: the near subset $O_{near}^t$, the mid subset $O_{mid}^t$, and the far subset $O_{far}^t$. Likewise, the $\mathbb{T}^{t-1}$ is also divided into three corresponding subsets $\mathbb{T}_{near}^{t-1}$, $\mathbb{T}_{mid}^{t-1}$, and $\mathbb{T}_{far}^{t-1}$. Considering that the nearer objects present less difficulty in matching, $O_{near}^t$ is prioritized for matching with $\mathbb{T}_{near}^{t-1}$ of the same distance level, and similarly, this process is followed for the other two levels. It is important to note that the unmatched objects and tracks within each level are transferred to the next level sets for subsequent matching.

## 3.3 Short-Long Terms Association

The associations can be categorized based on different timescales and degrees of occlusion: (i) the short-term association exists between adjacent frames, affected by partial occlusions, and (ii) the long-term association faces the challenge of re-associating objects that have been completely occluded and lost over some time. Thus, we propose *Short-Long Terms Association* (SLTA), which comprises *Short-term Association* and *Long-term Association*.

*Short-term Association.* The traditional association methods utilize the Kalman filter for motion prediction and track updates, employing IoU to calculate the correlation between predicted tracks and objects. We introduce *Pseudo-3D Adaptive Kalman Filter* (AKF) and *Distance-weighted IoU* (DIoU) to incorporate distance information as an auxiliary cue, enhancing the robustness of the association.

The track's state of the $t$-$1^{th}$ frame $\tau_i^{t-1} \in \mathbb{R}^{10}$ is represented as $(x, y, a, h, d, \delta)$, where $\delta$ is the motion velocity $(\dot{x}, \dot{y}, \dot{a}, \dot{h}, \dot{d})$ between the $t$-$1^{th}$ and $t$-$2^{th}$ frame. Our AKF assumes a constant velocity model with Gaussian noise. Firstly, we scale the prediction noise by the inverse distance [17], which leads to a smoother tracklet for tracks that are far away. We define the motion prediction step as:

$$\hat{\mu}_i^t = \mathbf{F}\mu_i^{t-1}, \tag{9}$$

$$\hat{\Sigma}_i^t = \mathbf{F}\Sigma_i^{t-1}\mathbf{F}^T + \frac{\mathbf{Q}}{d_{\tau_i^{t-1}}}, \tag{10}$$

where $\mu_i^{t-1}$ and $\hat{\mu}_i^t$ are the state means of $\tau_i^{t-1}$ and predicted track $\hat{\tau}_i^t$, respectively. The matrix $\mathbf{F}$ is the state transition matrix. The matrix $\Sigma_i^{t-1}$ and $\hat{\Sigma}_i^t$ are the state covariance at the $t$-$1^{th}$ frame and the $t^{th}$ frame. The matrix $\mathbf{Q}$ is the prediction noise covariance.

Besides, utilizing our $s_{oc}$ to adjust the measurement noise covariance allows some low-quality objects to use the predicted tracks more for state correction. We define the track update step as:

$$\mu_i^t = \hat{\mu}_i^t + K_i^t(\theta_i^t - \mathbf{H}\hat{\mu}_i^t), \tag{11}$$

$$K_i^t = \frac{\hat{\Sigma}_i^t \mathbf{H}^T}{\mathbf{H}\hat{\Sigma}_i^t \mathbf{H}^T + \mathbf{R}(1 - s_{oc})}, \tag{12}$$

where $\mu_i^t$ is the state mean of the updated track $\tau_i^t$, $\theta_i^t$ is the state mean of the object. The matrix $\mathbf{H}$ is the state transition matrix. $K_i^t$ is Kalman gain, reflecting the proximity of the updated track to the object. The matrix $\mathbf{R}$ is the measurement noise covariance.

Due to the strikingly similar IoU metrics between the predicted track and densely distributed objects, our DIoU incorporates distance to the vanilla IoU, thus providing highly reliable and discriminative correlations for association. Our DIoU is based on Generalized IoU (GIoU) [30] which is adjusted to [0,2]. We normalize the absolute difference in distance $\Delta d$ between the predicted track and the object as a weight. When $\Delta d = 0$, meaning they have the same distance, DIoU degenerates into GIoU. As $\Delta d$ gradually increases, GIoU can progressively be penalized, based on the prior that the same object should be proximate between adjacent frames in the distance dimension. The DIoU is expressed as follows:

$$\mathbf{DIoU} = \left\{ 1 - \left[ \frac{\Delta d}{\Delta d_{thr}} \cdot H(\Delta d_{thr} - \Delta d) \right]^\beta \right\} \mathbf{GIoU}, \tag{13}$$

where $\Delta d_{thr}$ is the distance association threshold, and when $\Delta d$ exceeds $\Delta d_{thr}$, it is considered that association will not occur. $\beta$ is used to adjust the penalty strength. $H(x)$ is the Heaviside function.

*Long-term Association.* Previous works utilize either state prediction by heuristic motion models (e.g. Kalman filters) with appearance features extracted by ReID models as the basis for re-association following long-term object loss. However, the former struggles to predict complex motions, and the latter faces challenges in learning stable features across different postures and viewpoints. In our *Long-term Association*, inspired by [32], we introduce a *Probabilistic Autoregressive Motion Model* (PAM) to learn reliable long-term motion prediction from the historical tracklets.

For a tracklet $\mathcal{T}_{i\{s:l\}}$ that is lost at $l^{th}$ frames, the long-term motion model essentially iterates to generate the predicted track for the current frame. Given the stochastic nature of object motion, generating a single deterministic predicted track is unreasonable. Therefore, multimodal stochastic modeling is required to compute the probability distribution of the next plausible predicted track, from which samples can be drawn. PAM also learns the interaction representation $\psi_i^{t-1}$ between the object and $N_\psi$ neighboring tracklets with MA-Net. The $\mathcal{T}_{i\{s:l\}}$ and $\psi_i^{t-1}$ are taken as inputs.

Notably, we lift the spatial dimensionality to preserve the role of distance information in re-association. For each track motion velocity in $(\dot{x}, \dot{y}, \dot{a}, \dot{h}, \dot{d})$, we utilize non-parametric k-means clustering to obtain K clusters and treat each cluster centroid as a discrete motion class. Through the PAM, the probability distribution of the predicted track can be obtained, as shown in the following equation:

$$P_i^t = \prod_{\xi \in \{x,y,a,h,d\}} p(\dot{\xi}_i^t | z_i^{t-2}, \delta_i^{t-1}, \psi_i^{t-1}), \tag{14}$$

where $z_i^{t-2}$ is the hidden state which carries all previous tracklet information until the $t$-$2^{th}$ frame. $\delta_i^{t-1}$ is the track motion velocity in the $t$-$1^{th}$ frame. The predicted probability distribution is obtained by iteration starting from the lost $l^{th}$ frame. By polynomial sampling, we can get $N_s$ possible predicted tracks. Additionally, due to the uncertainty in long-term motion prediction, we also incorporate an independent threshold constraint on the GIoU term in DIoU.

**Table 1: Comparison of the state-of-the-art methods under the "private detector" protocol on MOT17 test set. For each metric, the best is bolded and the second best is underlined. ↑ / ↓ indicates the higher/lower values denote the better performance.**

| Method | Source | Motion-only | HOTA↑ | MOTA↑ | IDF1↑ | AssA↑ | DetA↑ | FP↓ | FN↓ | IDs↓ | FPS↑ |
|---|---|---|---|---|---|---|---|---|---|---|---|
| Quo Vadis [6] | NeurIPS'22 | ✗ | 63.1 | 80.3 | 77.7 | 62.1 | 64.6 | 25,491 | 83,721 | 2,103 | 3.6 |
| SAT [36] | ACM MM'22 | ✗ | 64.4 | 80.0 | 79.8 | 64.4 | 64.8 | 25,125 | 86,505 | 1,356 | - |
| FineTrack [28] | CVPR'23 | ✗ | 64.3 | 80.0 | 79.5 | 64.5 | 64.5 | 21,750 | 90,096 | 1,272 | **35.5** |
| DiffMOT [21] | CVPR'24 | ✗ | 64.5 | 79.8 | 79.3 | 64.6 | 64.7 | 28,281 | 83,562 | 2,238 | - |
| ByteTrack [43] | ECCV'22 | ✓ | 63.1 | 80.3 | 77.3 | 62.0 | 64.5 | 25,491 | 83,721 | 2,196 | 29.6 |
| DNMOT [9] | ACM MM'23 | ✓ | 58.0 | 75.6 | 68.1 | - | - | 24,960 | 110,064 | 2,529 | - |
| OCSORT [3] | CVPR'23 | ✓ | 63.2 | 78.0 | 77.5 | 63.4 | 63.2 | **15,129** | 107,055 | 1,950 | 29.0 |
| MotionTrack [26] | CVPR'23 | ✓ | 65.1 | 81.1 | 80.1 | 65.1 | 65.4 | 23,802 | 81,660 | **1,140** | 15.7 |
| Hybrid [41] | AAAI'24 | ✓ | 63.6 | 79.3 | 78.4 | 63.2 | - | 35,4− | 79,1− | 2,109 | - |
| **DLT** | - | ✓ | **66.4** | **82.2** | **81.6** | **66.7** | **66.4** | 21,750 | **77,475** | 1,446 | 31.1 |

**Table 2: Comparison of the state-of-the-art methods on DanceTrack test set. For each metric, the best is bolded and the second best is underlined. ↑ / ↓ indicates the higher/lower values denote the better performance.**

| Method | Source | Motion-only | HOTA↑ | IDF1↑ | AssA↑ |
|---|---|---|---|---|---|
| FineTrack [28] | CVPR'23 | ✗ | 52.7 | 59.8 | 38.5 |
| DiffMOT [21] | CVPR'24 | ✗ | 62.3 | 63.0 | 47.2 |
| ByteTrack [43] | ECCV'22 | ✓ | 47.3 | 52.5 | 31.4 |
| DNMOT [9] | ACM MM'23 | ✓ | 53.5 | 49.7 | - |
| OCSORT [3] | CVPR'23 | ✓ | 54.6 | 54.6 | 40.2 |
| Hybrid [41] | AAAI'24 | ✓ | 62.2 | 63.0 | - |
| **DLT** | - | ✓ | **66.5** | **68.5** | **54.1** |

## 4 Experiments

Our novel DLT framework is evaluated on three key datasets: MOT17 [24], MOT20 [5], and DanceTrack [35]. Moreover, extensive ablation studies are conducted on MOT17 [24] and MOTSynth [8].

### 4.1 Setting

*Datasets.* MOT17 [24] and MOT20 [5], encompassing diverse real-world challenges like dense crowds and occlusions, require results to be uploaded to the motchallenge website for evaluation metrics. For ablation studies, we split MOT17 [24] into train and val sets. DanceTrack, a large-scale dataset known for body occlusions and random motion, is ideal for evaluating our motion-only DLT method. MOTSynth [8] is a large synthetic dataset providing authentic 3D coordinates for MDE training. We perform appropriate pre-processing to better tailor it to distance estimation tasks.

*Metrics.* Adhering to established MOT evaluation protocols, we use CLEAR metrics [1] including Multiple-Object Tracking Accuracy (MOTA), False Positive (FP), False Negative (FN), ID Switch (IDs), and Tracker Speed (FPS) to evaluate different aspects of tracking performance. In addition, we combine High Order Tracking Accuracy (HOTA) metrics with Detection Accuracy (DetA) and Association Accuracy (AssA) [20]. MOTA focuses on detection performance, IDF1 [31] emphasizes association performance, and HOTA offers a balanced measure of detection and association.

*Implementation details.* We train YOLOX [10] and our MDE on 4 NVIDIA Tesla A100 GPUs. The training approach for YOLOX is kept consistent with ByteTrack [43] to ensure a fair comparison. Our MDE is trained on MOTSynth using 6-frame video clips at 1280 × 720, with clips sampled with a uniform stride of 6 frames. We utilize a batch size of 4, Adam optimizer with an initial learning rate of $2 \times 10^{-5}$ for 50 epochs. For SDHM, $\alpha$ is 0.5. For SLTA, $\Delta d_{thr}$ is set at 5.0 with a $\beta$ of 2.0, and $N_s$ is set at 7.

### 4.2 Benchmark Evaluation

For a fair comparison, we classify methods into two types: those combining motion and appearance, and those only using motion.

*MOT17.* Despite not leveraging any appearance information, our DLT ranks first in most metrics on MOT17, indicating superior performance as shown in Table 1. By integrating distance cues, DLT achieves discrete and ordered matching with the SDHM strategy. It substantially surpasses the second-ranked MotionTrack [26] in both comprehensive metrics (+1.3 HOTA, +1.1 MOTA) and association metrics (+1.5 IDF1, +1.6 AssA). SDHM also notably reduces the FN metric by incorporating more objects into matching.

*MOT20.* DLT achieves advanced performance in most metrics on MOT20, noted for denser crowds and longer occlusions, as indicated in Table 3. It demonstrates robust generalization capabilities, markedly outperforming MotionTrack [26], which performs sub-optimally on MOT17 (+1.5 HOTA, +2.2 IDF1, +2.5 AssA). It also surpasses the second-ranked FineTrack [28] in comprehensive metrics (+0.7 HOTA, +0.4 MOTA). LSTA enhances the association between adjacent frames and re-association after long-term losses. Unlike FineTrack [28], DLT exhibits comparable performance in association metrics without complex appearance modules.

*DanceTrack.* Tracking in pseudo-3D space with distance cue, DLT demonstrates exceptionally strong performance on DanceTrack with more reliable matching and association, as shown in Table 2. Focusing on motion, especially with similarly dressed individuals, DLT significantly outperforms FineTrack [28] (+13.8 HOTA, +8.7 IDF1, +15.6 AssA). Additionally, DLT notably surpasses the second-ranked DiffMOT [21] (+4.2 HOTA, +5.5 IDF1, +6.9 AssA), showcasing its effectiveness in motion-based tracking.

**Table 3: Comparison of the state-of-the-art methods under the "private detector" protocol on MOT20 test set. For each metric, the best is bolded and the second best is underlined. ↑ / ↓ indicates the higher/lower values denote the better performance.**

| Method | Source | Motion-only | HOTA↑ | MOTA↑ | IDF1↑ | AssA↑ | DetA↑ | FP↓ | FN↓ | IDs↓ | FPS↑ |
|---|---|---|---|---|---|---|---|---|---|---|---|
| Quo Vadis [6] | NeurIPS'22 | ✗ | 61.5 | 77.8 | 75.7 | 60.1 | 63.8 | 26,249 | 87,594 | 1,187 | 2.2 |
| SAT [36] | ACM MM'22 | ✗ | 62.6 | 75.0 | 76.6 | 63.2 | 62.1 | **15,549** | 113,136 | 1,104 | - |
| FineTrack [28] | CVPR'23 | ✗ | 63.6 | 77.9 | **79.0** | 63.8 | 63.6 | 24,439 | 89,012 | 980 | 9.0 |
| DiffMOT [21] | CVPR'24 | ✗ | 61.7 | 76.7 | 74.9 | 60.5 | 63.2 | 27,217 | 91,804 | 1,509 | - |
| ByteTrack [43] | ECCV'22 | ✓ | 61.3 | 77.8 | 75.2 | 59.6 | 63.4 | 26,249 | 87,594 | 1,223 | 17.5 |
| DNMOT [9] | ACM MM'23 | ✓ | 58.6 | 70.5 | 73.2 | - | - | 29,314 | 122,252 | 987 | - |
| OCSORT [3] | CVPR'23 | ✓ | 62.4 | 75.7 | 76.3 | 62.5 | 62.4 | 19,067 | 105,894 | **942** | **18.7** |
| MotionTrack [26] | CVPR'23 | ✓ | 62.8 | 78.0 | 76.5 | 61.8 | 64.0 | 28,629 | **84,152** | 1,165 | 9.0 |
| Hybrid [41] | AAAI'24 | ✓ | 62.5 | 76.4 | 76.2 | 62.0 | - | 35,9− | 85,0− | 1,300 | - |
| **DLT** | - | ✓ | **64.3** | **78.3** | 78.7 | **64.3** | **64.5** | 22,672 | 88,617 | 1,133 | 18.2 |

**Table 4: Component analysis on the MOT17 val set.**

| SDHM | | SLTA | | HOTA↑ | MOTA↑ | IDF1↑ | AssA↑ | IDs↓ |
|---|---|---|---|---|---|---|---|---|
| SH | DH | SA | LA | | | | | |
| ✗ | ✗ | ✗ | ✗ | 67.7 | 76.6 | 81.2 | 71.6 | 523 |
| ✓ | ✗ | ✗ | ✗ | 69.0 (+1.3) | 77.5 (+0.9) | 81.9 (+0.7) | 72.1 | 513 |
| ✓ | ✓ | ✗ | ✗ | 69.5 (+0.5) | 78.1 (+0.6) | 82.2 (+0.3) | 72.5 | 497 |
| ✓ | ✓ | ✓ | ✗ | 70.2 (+0.7) | 78.5 (+0.4) | 82.9 (+0.7) | 73.1 | 482 |
| ✓ | ✓ | ✓ | ✓ | 70.5 (+0.3) | 78.7 (+0.2) | 83.3 (+0.4) | 73.4 | 462 |

**Table 5: Comparison of ST-Mamba and BUFF settings on the MOTSynth val set.**

| Setting | RMSE↓ | $\delta_{<1.25}$↑ | ALP$_{@0.5m}$↑ | ALP$_{@1m}$↑ | ALP$_{@2m}$↑ |
|---|---|---|---|---|---|
| Time-First | 1.9 | 97.1% | 49.2% | 67.4% | 88.2% |
| Space-Time | 1.7 | 97.8% | 49.9% | 69.9% | 89.8% |
| Top-Down | 1.7 | 97.9% | 52.8% | 69.6% | 89.7% |
| Down-Top | 1.8 | 97.3% | 51.7% | 68.2% | 88.9% |
| MDE | 1.7 | 98.2% | 53.4% | 70.1% | 90.2% |

## 4.3 Ablation Study

We conduct comprehensive ablation experiments on MOT17 and MOTSynth val sets.

*Component analysis.* We conduct an ablation study on the MOT17 val set to assess the contribution of each proposed component on our distance-aware tracking, as indicated in Table 4. SDHM is the overall matching strategy comprising *Score-based Hierarchizing* (SH) and *Distance-based Hierarchizing* (DH). SH notably enhances comprehensive metrics (+1.3 HOTA, +0.9 MOTA), while DH further improves performance (+0.5 HOTA, +0.6 MOTA). SLTA is a specific association scheme that includes *Short-term Association* (SA) and *Long-term Association* (LA). SA considerably boosts association capabilities (+0.7 IDF1, +0.6 AssA), while LA effectively resolves re-association problems after extended occlusions (-20 IDs).

*Analysis of distance estimators.* We report Root Mean Squared Error (RMSE) [18], $\tau$-Accuracy ($\delta$) [40], and Average Localization Precision (ALP) [45] as distance estimation evaluation metrics, as shown in Table 5. The ST-Mamba has different bidirectional 3D scans: a) Time-First, organizing spatial tokens by location then stacking them frame by frame; b) Space-First, arranging temporal tokens based on the frame then stacks along the spatial dimension; c) Space-Time, a hybrid of both Space-First and Time-First. The Space-First bidirectional scan is the most effective method which is employed in MDE. The BUFF employs bidirectional fusion to achieve multi-granularity extraction of spatiotemporal information. The experimental results demonstrate that bidirectional fusion offers advantages over both top-down and bottom-up fusion.

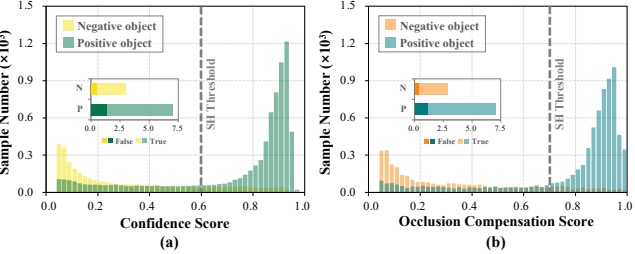

**Figure 5: Comparison of $s$ and $s_{oc}$ in SH on the MOT17 val set. It presents the statistical results of score distributions for both negative and positive objects.**

*Analysis of SDHM.* We perform detailed analyses on Score-based Hierarchizing (SH) and Distance-based Hierarchizing (DH).

(i) Effect of SH. SH aims to separate negative and positive objects based on object scores, avoiding the generation of erroneous associations. We introduce an *Occlusion Compensation Score* ($s_{oc}$) to counter the impact of occlusions on scoring, adjusted by approximate visibility ratios obtained from the distance order. By randomly sampling $10^4$ object samples, we statistically analyze the distribution of both negative and positive object scores. The SH threshold divides objects into low- and high-score subsets. The error rate is the ratio of negative objects in the high-score subset to the total negatives, or the ratio of positive objects in the high-score subset to the total positives. Compared to the confidence score ($s$) directly provided by the detector, our $s_{oc}$ plays an important role in improving hierarchizing accuracy, as illustrated in Figure 5.

**Table 6: Comparison of DH settings on the MOT17 val set.**

| Setting | HOTA↑ | MOTA↑ | IDF1↑ | AssA↑ | IDs↓ |
|---|---|---|---|---|---|
| DH | 69.5 | 78.1 | 82.2 | 72.5 | 497 |
| DH w/ EN | 69.4 (-0.1) | 77.8 (-0.3) | 82.1 | 72.4 | 502 |
| DH w/ F→N | 69.2 (-0.3) | 77.7 (-0.4) | 81.9 | 72.2 | 510 |
| DH w/ L | 69.1 (-0.4) | 77.8 (-0.3) | 82.2 | 72.2 | 512 |
| DH w/ H+L | 69.6 (+0.1) | 77.9 (-0.2) | 82.3 | 72.4 | 492 |

**Table 7: Comparison of LA methods on the MOT17 val set.**

| Setting | HOTA↑ | MOTA↑ | IDF1↑ | AssA↑ | IDs↓ |
|---|---|---|---|---|---|
| Static | 66.4 | 75.6 | 79.3 | 72.1 | 575 |
| AKF | 69.5 | 77.3 | 82.1 | 72.8 | 518 |
| PAM | 70.5 | 78.7 | 83.4 | 73.4 | 462 |
| PAM w/o MA | 70.4 | 78.2 | 83.0 (-0.4) | 73.3 | 472 (+10) |
| PAM w/o SPL | 70.2 | 77.7 | 82.7 (-0.7) | 73.0 | 485 (+23) |

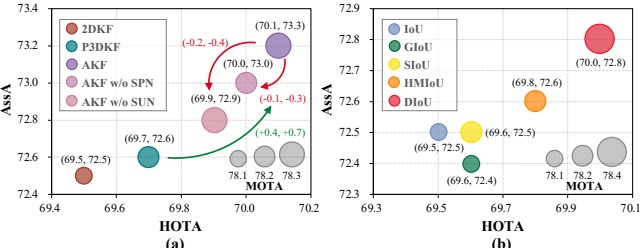

**Figure 6: Comparison of Kalman filter and IoU variants in SA on the MOT17 val set.**

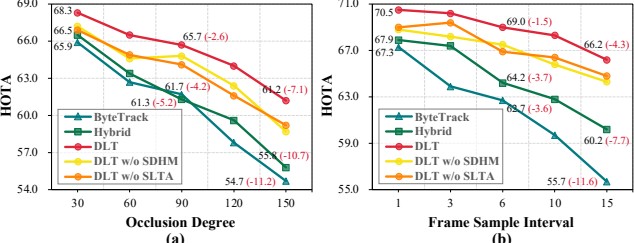

**Figure 7: Comparison of the influence of occlusion degree and low frame rate on the MOT17 val set.**

(ii) Effect of DH. Integrating DH into the high-score subset of SH aims for discrete and ordered matching in distance dimension. As shown in Table 6, we explore several various DH settings. Switching from equal-distance decomposition (ED) to equal-object-number decomposition (EN) decreases performance slightly (-0.1 HOTA, -0.3 MOTA). Matching from far to near (F→N) is obviously less effective than near to far (N→F) (-0.3 HOTA, -0.4 MOTA). Using DH in both high-score (H) and low-score (L) subsets shows essentially equivalent performance (+0.1 HOTA, -0,2 MOTA), unlike its negative impact when applied only in L (-0.4 HOTA, -0.3 MOTA).

*Analysis of SLTA.* We perform experimental analyses of Short-term Association (SA) including *Pseudo-3D Adaptive Kalman Filter* (AKF) and *Distance-weighted IoU* (DIoU), and Long-term Association (LA) including *Probabilistic Autoregressive Motion Model* (PAM).

(i) Effect of SA. Our SA integrates AKF and DIoU for accurate state estimation and track updates. AKF evolves from 2DKF [2] to P3DKF, and utilizes distance for scaling prediction noise (SPN) and update noise (SUN) with our $s_{oc}$, enabling adaptive noise adjustments. As shown in Figure 6 (a), compared to 2DKF, AKF introduces a distance dimension to state estimation, with SPN and SUN further enhancing association performance (+0.4 HOTA, +0.7 AssA). Furthermore, we develop a DIoU, integrating object-level distance as a potential cue into the correlation calculation, outperforms IoU, GIoU [30], SIoU [11], and HMIoU [41], as indicated in Figure 6 (b).

(ii) Effect of LA. Our LA tackles extreme occlusion, thereby allowing up to 120 frames for object loss to demonstrate the capability of handling long-term tracking, as shown in Table 7. We evaluate a no-motion model (assuming the lost object remains static), a linear model (i.e., AKF), and a nonlinear multimodal model (i.e., our PAM) for long-term identity retention. Compared to Static and AKF, PAM remarkably addresses long-term occlusions by capturing the randomness and multimodality of human motion. Additionally, MA-Net's consideration of neighboring object interactions boosts

association accuracy (+0.4 IDF1, -10 IDs). PAM also employs polynomial sampling (SPL) to predict multiple potential movement areas, outperforming the single top-1 area prediction (+0.7 IDF1, -23 IDs).

*Extra evaluation of occlusion degree.* MOT17 provides the visibility ratio annotations of objects, and we consider those with a visibility ratio below 0.3 as occluded. The occlusion degree can be represented by the number of frames in which the occlusion state persists. As the occlusion degree increases, DLT shows slower declines (-7.1 HOTA) compared to ByteTrack [43] (-11.2 HOTA) and Hybrid [41] (-10.7 HOTA), as indicated in Figure 7 (a).

*Extra evaluation of low frame rates.* At low frame rates, the impact of occlusion is further magnified. We use a frame sample operation for simulation as shown in Figure 7 (b). The performance of ByteTrack [43] and Hybrid [41] declines rapidly with decreasing frame rates (-11.6 HOTA and -7.7 HOTA), whereas DLT demonstrates superior low-frame-rate stability (-4.3 HOTA). In addition, SDHM and SLTA play obvious roles in performance retention.

## 5 Conclusion

In this paper, we propose an innovative "Detecting-Lifting-Tracking" (DLT) framework for 2D MOT. We introduce a Mamba Distance Estimator, incorporating historical information to mitigate temporary occlusions, achieving object-level pseudo-3D lifting. To realize distance-aware tracking, we propose a Score-Distance Hierarchical Matching strategy for discrete and sequential matching, along with a Short-Long Terms Association scheme to address issues of dense crowds in short-term associations between adjacent frames and severe occlusions in long-term re-associations. Extensive benchmark validations and ablation studies demonstrate the superior performance of our DLT, showcasing the enormous potential to overcome occlusion challenges. We hope to inspire future research to further explore enhancing 2D MOT tasks with 3D information.

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
