# OpenReview forum: "Object-Level Pseudo-3D Lifting for Distance-Aware Tracking"
_acmmm.org/ACMMM/2024/Conference — MM2024 Poster_

### Official Review · Reviewer_jKsB · 2024-05-11

**Rating:** 4
**Confidence:** 2

**Summary:**

This paper introduces an innovative 2D MOT framework, “Detecting-Lifting-Tracking” (DLT). Specifically, a mamba distance estimator is proposed to obtain the distances from objects to a monocular camera, achieving object-level pseudo-3D lifting. Then, a score-distance hierarchical matching technique is introduced to achieve discrete and sequential matching. Finally, a short-long term association strategy is employed to optimize short-term associations between adjacent frames and to enhance the long-term re-association capability. A large number of experiments demonstrate the effectiveness of the proposed method.

**Strengths:**

1. The paper is well written and can be easily understood by the reader.
2. The idea of the paper is interesting, presenting a novel Mamba distance estimator that combines historical information for time-consistent distance estimation, enabling object-level pseudo-3D lifting.

**Limitations:**

- The necessity of mamba. Whether the performance of the proposed algorithm will be improved if the mamba operator is replaced by using transformer.

**Suitability:**

2

---

### Official Review · Reviewer_SY2p · 2024-05-23

**Rating:** 4
**Confidence:** 3

**Summary:**

This paper presents Detecting-Lifting-Tracking (DLT) framework for Multi-Object Tracking (MOT). DLT firstly performs Object-Level Distance Estimation (OLDE) that estimates the distance between each object in a frame and the camera using Mamba Distance Estimator (MDE). The estimated object-level distances are then used in the distance-aware tracking process consisting of two subprocesses, Score-Distance Hierarchical Matching (SDHM) and Short-Long Terms Association (SLTA). The experimental results on the three datasets show the effectiveness of the proposed DLT.

**Strengths:**

- DLT seems to apply (or extend) very new methods to each of its component processes like Mamba in OLDE, Score-/Distance-based Hierarchizing, GIoU in the short-term association and Probabilistic Autoregressive Motion (PAM) in the long-term association.
- It seems that very extensive experiments have been conducted to validate the effectiveness of each component process in DLT.

**Limitations:**

- I think the biggest problem is that this paper contains too many contents, so each component process in DLT is described very briefly. By referring to the supplemental material, I could manage to get a rough idea about each of the component processes and their relations. But, it seems very difficult to get reasonable understanding of DLT only from the main paper.
- Related to the above point, many things are described without enough explanation. For example, why is the distance $d$ of an object not directly estimated, that is, why are $\hat{\mu}$ and $\hat{\sigma}^2$ of a Gaussian distribution for distances are estimated? Why is GNLL is used (what is the advantage of GNLL)? What is meant by "hierarchical strategy sensitivity to occlusions"? What is the rationale behind Eq. (8) to compute $s_{oc}$? How are $\boldsymbol{F}$, $\boldsymbol{Q}$, $\boldsymbol{H}$, $\boldsymbol{R}$, $\boldsymbol{GIoU}$ computed? What is the rationale behind Eq. (13)? What is MA-Net? and so on. Despite these many unknown things, I could manage to imagine how DLT works by referring to the supplemental material.
- DLT seems to be a combination of the component processes each of which is a slight extension of an existing method. For instance, Mamba is used to extend OLDE to the time dimension, Hierarchizing, Adaptive Kalman Filter (AKF) and GIoU are extended by adopting object-level distances. So, someone may think the contributions of this paper are relatively small.
- Is it possible to devise a baseline method in order to examine how important object-level distances?
- How can I understand Figure 5?

**Suitability:**

2

---

### Official Review · Reviewer_hcXE · 2024-05-25

**Rating:** 5
**Confidence:** 4

**Summary:**

The "Detecting-Lifting-Tracking" (DLT) framework for 2D multi-object tracking utilizes pseudo-3D lifting and distance-aware tracking, achieving state-of-the-art performance on MOT17, MOT20, and DanceTrack without relying on appearance cues.

**Strengths:**

1.. Lifting: Employing Object-Level Distance Estimation (OLDE) with the Mamba Distance Estimator to create pseudo-3D information for each object.
2.. Tracking: Implementing distance-aware tracking through Score-Distance Hierarchical Matching (SDHM) combined with Short-Long Terms Association (SLTA).
3. This paper is well-written.

**Limitations:**

1.	Is it suitable for real-time? Score-Distance Hierarchical Matching (SDHM) combined with Short-Long Terms Association (SLTA) are post-processing.
2.	The complexity of Detecting-Lifting-Tracking” (DLT) framework can be discussed since MDE can reduce some computation effort.
3.	Can you show the ablation studies for the Mamba module?
4.	The performance of DIoU should be testified.

**Suitability:**

3

---

### Meta-Review · Area_Chair_adE5 · 2024-06-28

**Recommendation:** Accept (Poster)
**Confidence:** 5

**Metareview:**

The paper is well written, and an interesting idea - a Mamba distance estimator that combines historical information for time-consistent distance estimation enabling object-level pseudo-3D lifting - is proposed. All the three reviewers recommend acceptance. AC agrees with the reviewers.